# Periodical Moving Average Accelerates Gradient Accumulation for Post-Training

## Abstract

High gradient variance challenges training Large Language Models (LLMs) on memory-limited devices. Existing practical approaches, such as small batch size or using Gradient Accumulation (GA), face the dilemma between low convergence rates due to high variance in parameter updates and long training times due to the serial GA process. In this paper, we identify that the exponential nature of the Exponential Moving Average (EMA) rapidly forgets historical gradients at an exponential rate in momentum updates, making it difficult to utilize the historical gradients to stabilize the update steps. To address this issue, we embed the idea of GA into the momentum update and propose the Periodical Moving Average (PMA) technique. PMA splits the training steps into periods and employs moving averages instead of EMA in each period. We apply PMA to AdamW and Lion, resulting in AdamW-PMA and Lion-PMA. Theoretical analysis demonstrates that AdamW-PMA achieves a comparable convergence rate with Adam. Extensive experiments showcase the superiority of PMA on post-training tasks, including Supervised Fine-Tuning and Direct Preference Optimization, that the PMA-based methods achieve approximately at least $2\times$ speedup and higher scores on downstream tasks.

## 1 Introduction

Scaling up Large Language Models (LLMs) has been empirically evaluated as a necessary approach to enhance their capabilities (Radford et al., 2019; Kaplan et al., 2020; Brown et al., 2020; Hoffmann et al., 2022; Zhang et al., 2022; Touvron et al., 2023a;b; Achiam et al., 2023; Bi et al., 2024). Each stage of the LLM post-training, including Supervised Fine-tuning (SFT), and reinforcement-learning-based training, including Reinforcement Learning from Human Feedback (RLHF) (Ouyang et al., 2022) and beyond (OpenAI, 2024), demands high computation costs on GPU-sufficient clusters (Lee & Sengupta, 2022). However, scaling up require larger GPU memories, posing a challenge for implementation on GPU-memory-limited devices.

Alternative approaches to training LLMs on GPU-memory-limited devices share a common weakness of prolonged training time consumption. The most straightforward method is to employ a small batch size. However, the large gradient noise slows down the training process and makes the model hard to converge. Another approach is to use Gradient Accumulation (GA), which involves multiple backpropagations followed by gradient averaging before a parameter update step, achieving an equivalent large batch size. This approach converts the utilization of abundant GPU resources in parallel processing into a sequential process, also albeit at the expense of increased training time.

In this paper, we propose *Periodical Moving Average* (`PMA`), a momentum update method designed to accelerate momentum-based optimizers in LLM training on GPU-memory-limited devices. The technical challenge lies in achieving both low variance and low time cost simultaneously. On the one hand, from the perspective of GA, it is difficult to take more parameter updates without interrupting the GA process or requiring extra memory allocation. On the other hand, from the side of small-batch training, stabilizing the parameters becomes challenging when gradients are sampled from a small batch, leading to increased variance. However, we observe that post-training usually uses a lower learning rate than pre-training, preventing the parameters after training from deviating significantly from the pre-trained model. This suggests that the gradients over the last few steps tend to have similar expectations. `PMA` divides the entire training process into multiple periods, each consisting of $K$ steps. During each period, the momentum is updated using a moving average instead of the

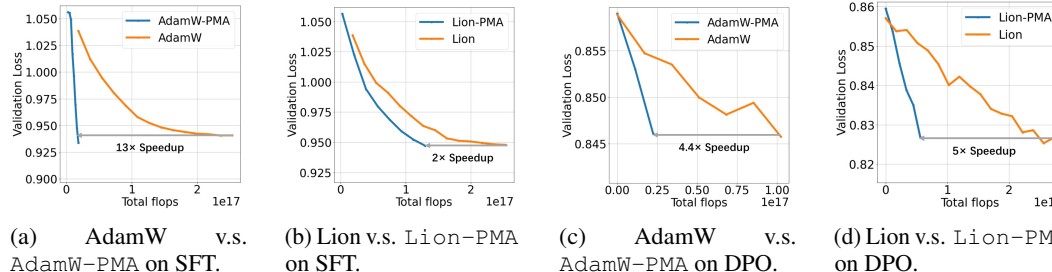

(a) AdamW v.s. `AdamW-PMA` on SFT.

(b) Lion v.s. `Lion-PMA` on SFT.

(c) AdamW v.s. `AdamW-PMA` on DPO.

(d) Lion v.s. `Lion-PMA` on DPO.

Figure 1: Optimizers with `PMA` achieves about $2\times$ speedup compared with the optimizers with EMA. (1a1b) Comparison of the number of steps needed to achieve the same validation loss with Phi-2 2.7B model on SFT task and Alpaca dataset. (1c1d) Comparison of the number of steps needed to achieve the same validation loss with Phi-2 2.7B model on DPO task and HH-RLHF dataset.

Exponential Moving Average (EMA) (Kingma & Ba, 2014; Loshchilov & Hutter, 2017; Chen et al., 2023; Liu et al., 2023), effectively reducing momentum variance. Between periods, the vanilla EMA is employed to leverage the high convergence rate of standard EMA-based optimizers.

The challenge of *trajectory deviation* arises when applying `PMA` to existing optimizers. Since the same weight is applied to gradients within a period, the averaged gradients may lead to a parameter update trajectory that deviates from the desired path. However, the optimizer cannot access the true expectations of the stochastic gradients, making it difficult to detect or regulate any potential deviation. To mitigate this trajectory deviation, we implement a linear decay of the learning rate within each period, while resetting the learning rate to its initial value at the start of each new period. This strategy ensures that the gradients do not deviate excessively from one another, thereby stabilizing the update steps between periods and maintaining the desired trajectory.

By applying `PMA`, we modify AdamW (Loshchilov & Hutter, 2017) and Lion (Chen et al., 2023), proposing `AdamW-PMA` and `Lion-PMA`. To verify the effectiveness of `AdamW-PMA` and `Lion-PMA`, we conduct extensive experiments covering the post-training process of an LLM, including Supervised Fine-training (SFT) and Direct Preference Optimization (DPO) (Rafailov et al., 2023) on GPT-2 (Brown et al., 2020), Phi-2 (Javaheripi et al., 2023), Qwen1.5 (Team, 2024a), Qwen2 (Yang et al., 2024) and Llama2 (Touvron et al., 2023b). Empirical evaluation shows that `AdamW-PMA` and `Lion-PMA` achieve approximately $2\times$ speedup in the post-training process and deliver better performance on downstream tasks. Furthermore, we provide a theoretical analysis of `AdamW-PMA` on the learning rate strategy and regret bound, demonstrating that the theoretical convergence properties of `AdamW-PMA` are on par with those of Adam.

Our technical contributions are summarized as follows:

- We propose Periodical Moving Average (`PMA`), a momentum update method to accelerate LLM fine-training on GPU-memory-limited devices. We adopt `PMA` to AdamW and Lion, to propose `AdamW-PMA` and `Lion-PMA`. Both algorithms stabilize the training and cost no more memory and computation overhead in each step, achieving the same level of loss with less time and less data compared to the original algorithms.

- We conduct extensive experiments across model sizes (from 0.1B to 7B) and training tasks (SFT and DPO) to evaluate the performance of `AdamW-PMA` and `Lion-PMA`. `PMA`-modified methods achieve approximately $2\times$ speedup in the training process and deliver better performance on downstream tasks.

- We provide theoretical analyses of `AdamW-PMA`. The regret analysis on convex functions shows that the theoretical convergence property of `AdamW-PMA` is at the same level as Adam. The convergence analysis of the small update steps shows the correctness of our designed learning rate strategy.

## 2 PRELIMINARIES

### 2.1 BACKGROUND: FIRST ORDER OPTIMIZATION

Adam (Kingma & Ba, 2014; Reddi et al., 2019) and AdamW (Loshchilov & Hutter, 2017) amalgamate adaptive and momentum-based methods, emerging as widely adopted optimizers for LLM training. Adam's update rule is as follows: Given an objective function $f : \mathbb{R}^d \to \mathbb{R}$, at time $t$, after computing the gradient $g_t = \nabla f$, the first and second momenta are updated as $m_t = \beta_1 m_{t-1} + (1 - \beta_1)g_t$ and $v_t = \beta_2 v_{t-1} + (1 - \beta_2)g_t^2 + \epsilon$, respectively, where $\epsilon$ is a small constant. Each momentum is then debiased by dividing by $\sqrt{1 - \beta_1^t}$ and $\sqrt{1 - \beta_2^t}$ to obtain $\hat{m}_t$ and $\hat{v}_t$, respectively. Since the weight of the historical gradients decays exponentially, this method is also known as the EMA. Finally, the model parameter is updated by $x_{t+1} \leftarrow x_t - \gamma \cdot \frac{\hat{m}_t}{\sqrt{\hat{v}_t}}$, where $\gamma$ represents the learning rate, and all operations are element-wise. Recent representative advances by Chen et al. (2023) is Lion, which employs with the first momentum only, and uses EMA to update it.

However, Adam-based methods face significant challenges related to high gradient variance in memory-limited environments. Language model training inherently presents a high-variance optimization problem (McCandlish et al., 2018). To tackle this issue, high-performance clusters are often employed to increase the batch size (Touvron et al., 2023a). Conversely, reducing the batch size exacerbates stochastic gradient noise, impeding model convergence (Yuan et al., 2016; Bottou et al., 2018; Kunstner et al., 2023; Fu et al., 2023).

### 2.2 THEORETICAL SOLUTION: VARIANCE REDUCTION IN SGD

In this subsection, we discuss the methods to reduce gradient variance in SGD (Bottou et al., 2018) and their drawbacks. The gradient aggregation methods reduce variance by reusing previously computed information. Specifically, at time step $t$, SVRG (Johnson & Zhang, 2013) maintains a copy of the historical parameter $\theta_k$ where $k < t$. The iterate averaging method (Polyak, 1991) stores the parameters after each SGD step and returns the average of the stored parameters. Recent advances proposed methods with recursive gradient updates without storing past gradients, such as SARAH (Nguyen et al., 2017) and STORM (Cutkosky & Orabona, 2019).

However, current variance reduction methods either require large memory space, which is not feasible when tuning the LLMs, or have low sampling efficiency. SAGA needs to store a gradient for each data sample, and the memory cost is proportional to the size of the dataset. The iterate averaging method needs to store all the updated parameters, with memory proportional to the number of steps. SVRG needs to sample a large batch in each step to reduce the gradient variance. Although SARAH and STORM does not require storing past gradients, they need more than one more back-propagation on the past parameters for one update, leading to a high computation overhead for training LLMs.

### 2.3 PRACTICAL SOLUTION: GRADIENT ACCUMULATION

To mitigate the high variance of stochastic gradients in memory-limited scenarios, a straightforward alternative method is GA[1]. GA involves partitioning the large batch into $K$ smaller batches and computing the gradient on each small batch without overwhelming the available memory. Subsequently, the gradients from these small batches are averaged to obtain the gradient of the large batch for parameter update. Importantly, the accumulated gradient obtained through GA remains equivalent to the gradient obtained directly from the large batch.

However, GA faces significant practical challenges, primarily related to computational time. This is due to the fact that GA mitigates memory overhead by transforming parallel computations, typical in large clusters, into serial computations on memory-limited devices. Specifically, GA requires performing feed-forward and back-propagation $K$ times before updating the parameters once, resulting in lower computational efficiency. Pham et al. (2023) modified the GA process to reduce the memory cost, but they did not touch the target of speeding up training on memory-limited devices.

---

[1]We acknowledge that applying memory-efficient optimizers, such as Shazeer & Stern (2018); Luo et al. (2023); Zhao et al. (2024); Zhang et al. (2024), may also be a practical approach. However, we claim that this approach is not as practical as GA in our scenario. Due to the page limit, the corresponding discussion is presented in Appendix A.

## 2.4 THE DILEMMA

Throughout the analysis in Section 2.2 and Section 2.3, we identified a dilemma between theoretical solutions and practical approaches when performing high-variance optimization on GPU-memory-limited devices. Theoretical variance-reduction solutions often entail high memory overhead, which is impractical in our scenario. Existing practical approaches either suffer from slow convergence rates due to high gradient variance or prolonged training times due to serial operations of GA. Since the crux of this dilemma lies in the high variance of the gradients, it is imperative to design a method to reduce the variance of parameter updates while maintaining a similar level of memory and computational costs as the current approaches.

## 3 METHODOLOGY: PERIODICAL MOVING AVERAGE

To relieve the dilemma in Sec.2.4, we introduce PMA as an extension to EMA to reduce the variance of the gradients. Section 3.1 specifies the high-level design idea of PMA and explains the connection and difference and existing work, providing an intuitive explanation. Section 3.2 dives into the design details by introducing the dynamics of $\beta$ and the learning rate schedule. Section 3.3 provides two cases, demonstrating how PMA can be applied to AdamW and Lion.

### 3.1 HIGH-LEVEL IDEA

**GA reduces variance, let's mimic it.** At a high level, the PMA mimics the GA process in the momentum update. Unlike the EMA method, which forgets the historical gradient with an exponential rate, PMA retains the same weights for some recent gradients. This is realized by splitting the training process into periods and performing the vanilla moving average for momentum updates in each period. This approach mimics the GA process, thus ensuring that the momenta have approximately low variance compared with those of EMA-based optimizers with GA (Sec. 3.2.1).

**GA updates no parameters, let's update them.** The core concept of PMA revolves around taking small steps forward during each GA round. Specifically, the main procedure of PMA alternates between using large and small learning rates during GA. We refer to the step using a large learning rate as the *large update step*, while the others are termed *small update steps*. Each large update step, following $K$ small update steps, mimics an update step in EMA-based optimizers with GA. Conversely, each small update step, employing a small learning rate, aims to accelerate convergence while avoiding excessive movement to disrupt the GA process (Sec. 3.2.2).

The superiority of PMA over EMA-based optimizers with GA mainly stems from the update steps with the smaller learning rate. These small update steps, interspersed between two large update steps, allow the parameters to be updated after each round of gradient computation, rather than waiting until all $K$ rounds of gradients are accumulated.

### 3.2 DETAILED DESIGN

In this subsection, we introduce the update method of the first momentum as an example.

#### 3.2.1 MOMENTUM UPDATE: DYNAMICS OF BETAS

Instead of setting $\beta$ to be a constant during the training process as EMA, the adaptation of PMA employs dynamic $\beta$[2] to achieve the vanilla moving average nature of PMA. Since the weights of each historical gradient in a period is required to be the same, the $\beta$ should decay through a period, assigning a reducing $\beta$ to the newer gradient. In the following, we introduce the dynamics of $\beta$ in the large and small update steps separately, and an illustration of the $\beta$ is given in Fig. 2a.

**At Large Update Steps: Small Gradient Weight for Low Variance.** At the first small update step after a large update step, i.e., $\tau = 0$, the updates of the momentum is given by $m_t \leftarrow \beta_1 m_{t-1} + (1 - \beta_1) g_t / K$. This update method bears resemblance to that of EMA, wherein historical gradients decay

---

[2]$\beta$ is usually known as the weight of the momentum and $1 - \beta$ is the weight of the gradient. In the following text, we focus more on the weight of the gradient, while keep using the terminology of $\beta$ for simplicity.

with the factor $\beta$ and the current gradient is multiplied by $1 - \beta$. However, a key distinction from EMA lies in our approach of dividing the current $g_{t,0}$ by $K$, and after the update, the first and second momenta are scaled by $K$. This operation serves a dual purpose: firstly, to mitigate the variance of the first and second momenta, and secondly, to ensure that after the subsequent $K$ small steps, the accumulated gradients $g_{t,0}, \ldots, g_{t,K-1}$ are weighted equally in $m_{t,K-1}$, thereby aligning the momenta with those in GA.

**At Small Update Steps: Moving Average with Dynamic Weights.** When $\tau = 1, \ldots, K-1$, the update of momentum are given by $m_t \leftarrow \frac{\tau}{\tau+1} m_{t-1} + \frac{1-\beta_1}{\tau+1} g_t$. Initially, the current gradient is multiplied by $1 - \beta$. Our update method for the small steps replaces the EMA method used in RMSprop and Adam with the moving average method. This modification aims to emulate the GA method, ensuring that the gradients of the small update steps have the same weight in $m_{t,K-1}$, thereby enabling the large update step to mimic an update step in EMA-based optimizers while keeping a low variance. Notably, in $m_{t,K-1}$, $m_{t-1,K-1}$ has a weight of $1 - \beta_1$, while $g_{t,\tau}$ has a weight of $\frac{1-\beta_1}{K}$ for all $\tau$. These weights mirror those in GA.

---

**Algorithm 1:** AdamW-PMA

**input**:   $\gamma$(lr), $\beta_1$, $\beta_2$(betas), $\theta_0$(params),
      $f(\theta)$(objective), $\epsilon$(epsilon), $\lambda$(weight decay),
      $K$(accumulate iterations)

**Data:** $m_0 \leftarrow 0$, $v_0 \leftarrow 0$

1  **for** $t = 1 \rightarrow \ldots$ **do**
2      $g_t \leftarrow \nabla_\theta f_t(\theta_{t-1})$;
3      $\tau \leftarrow t \% K$;
4      **if** $\tau = 0$ *and* $t > 0$ **then**
         // For every $K$ steps, there is a large update step.
5         $\gamma_t \leftarrow \gamma$;
6         $m_t \leftarrow \beta_1 m_{t-1} + (1 - \beta_1) g_t / K$;
         // Divide gradient by $K$ for stability.
7         $v_t \leftarrow \beta_2 v_{t-1} + (1 - \beta_2) g_t^2 / K$;
8      **else**
         // For every $K$ steps, there is $K - 1$ small update steps.
9         $\gamma_t \leftarrow \gamma / \sqrt{K}$;     // Shrink the learning rate by $1/\sqrt{K}$.
10        $m_t \leftarrow \frac{\tau}{\tau+1} m_t + \frac{1-\beta_1}{\tau+1} g_t$;   // Moving average instead of EMA.
11        $v_t \leftarrow \frac{\tau}{\tau+1} v_t + \frac{1-\beta_2}{\tau+1} g_t^2$;
12      $\hat{m}_t \leftarrow m_t / (1 - \beta_1^{t//K})$; // Debias. "//" refers to division with remainder.
13      $\sqrt{\hat{v}_t} \leftarrow \sqrt{v_t / (1 - \beta_2^{t//K})} + \epsilon$;
14      $\hat{\theta}_t \leftarrow (1 - \gamma_t \lambda) \theta_{t-1}$;     // Weight decay.
15      $\theta_t = \hat{\theta}_t - \gamma_t \hat{m}_t / \sqrt{\hat{v}_t}$;     // Parameter update.
16      **if** $\tau = 0$ *and* $t > 0$ **then**
17        $\hat{m}_t \leftarrow K \hat{m}_t$;     // Rescale the momentum after large update step.
18        $\hat{v}_t \leftarrow K \hat{v}_t$;
19  **return** $\theta_t$;

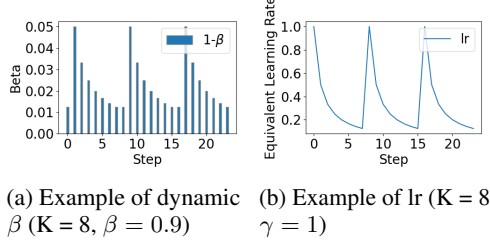

(a) Example of dynamic $\beta$ (K = 8, $\beta = 0.9$)

(b) Example of lr (K = 8, $\gamma = 1$)

Figure 2: Illustrations of the dynamics of $\beta$ and learning rate.

---

**Algorithm 2:** Lion-PMA

1  **for** $t = 1 \rightarrow \ldots$ **do**
2      $g_t \leftarrow \nabla_\theta f_t(\theta_{t-1})$;
3      $\tau \leftarrow t \% K$;
4      **if** $\tau = 0$ *and* $t > 0$ **then**
5        $\gamma_t \leftarrow \gamma$;
6        $u_t \leftarrow \beta_1 m_{t-1} + (1 - \beta_1) g_t / K$;
7        $u_t \leftarrow \text{sign}(u_t)$;
8      **else**
9        $\gamma_t \leftarrow \gamma / K$;
10       $u_t \leftarrow \frac{\tau}{\tau+1} m_t + \frac{1-\beta_1}{\tau+1} g_t$;
11       $u_t \leftarrow \text{sign}(u_t)$;
12       $m_t \leftarrow \frac{\tau}{\tau+1} v_t + \frac{1-\beta_2}{\tau+1} g_t^2$;
13      $\hat{\theta}_t \leftarrow (1 - \gamma_t \lambda) \theta_{t-1}$;
14      $\theta_t = \hat{\theta}_t - \gamma_t u_t$;
15      **if** $\tau = 0$ *and* $t > 0$ **then**
16        $\hat{m}_t \leftarrow K \hat{m}_t$;
17  **return** $\theta_t$;

### 3.2.2 THE LEARNING RATE SCHEDULE

To cope with the dynamics of the momentum update in Sec.3.2.1, we design a learning rate scheduler that differentiates between small and large update steps in this subsection. This strategy employs a decaying learning rate for small update steps instead of using a uniform learning rate for each step. The necessity of such a design stems from mimicking GA to avoid excessive movement that could disrupt the mimicked GA process. An example of the dynamic learning rate is illustrated in Fig. 2b.

The design intuition for the learning rate strategy aims to advance the $K$ small update steps relatively further to expedite convergence compared to Adam with GA while preventing these steps from

advancing too much and causing $m_{t,K}$ to deviate significantly from the momentum of EMA-based optimizers with GA. With this intention, we scale the momentum by $1/K$ at the small update steps after the momentum update. Considering the scaling of momenta in Lines 16-18 and the momentum update method with varying weights, the actual learning rate at small update step $\tau$ is $\eta \cdot \frac{K}{K} \cdot \prod_{i=1}^{\tau} \frac{i}{i+1} = \eta/(\tau + 1)$, decreasing at a linear rate.

### 3.3 CASE STUDY

#### 3.3.1 FROM ADAMW TO `AdamW-PMA`

We modify AdamW to `AdamW-PMA` by replacing the EMA with the PMA as introduced in Sec.3.2. The pseudo-code of `AdamW-PMA` is provided in Alg. 1. We replace the EMA method for the first and second momentum updates in AdamW with PMA, following Sec.3.2.1. Note that the update of AdamW is computed by $m/\sqrt{v}$ (ignoring weight decay), and both the first and second momentum are scaled by $K$ at large update steps. The learning rate is scaled by $1/K$ at small update steps, resulting in a learning rate that is effectively scaled by $\sqrt{K}/K = 1/\sqrt{K}$. For the remaining components of `AdamW-PMA`, we keep them unchanged from AdamW.

#### 3.3.2 FROM LION TO `Lion-PMA`

The modification to Lion follows a similar approach to AdamW. It is noteworthy that the learning rate is decayed by $1/K$ at small update steps, instead of $1/\sqrt{K}$ in `AdamW-PMA`. The reason is that there is no second momentum in Lion. Thus, we choose $1/K$ to align with the rescaling of momentum after the large update step, ensuring that the actual learning rate linearly decays as discussed in Sec.3.2.2. For the remaining components of `Lion-PMA`, we keep them unchanged from Lion. The pseudo-code of `Lion-PMA` is illustrated in Alg. 2.

## 4 THEORETICAL ANALYSIS

### 4.1 CONVERGENCE ANALYSIS

In this section, we provide a theoretical analysis on the convergence property of `AdamW-PMA`. Specifically, we focus on the convergence properties concerning the number of large update steps. This focus is due to the time cost between two large steps being approximately equal to the time between two updates of Adam with GA. During the analysis, we slightly modify the notations for ease of analysis. Unlike Alg. 1, where the index of small update steps ranges from 0 to $K-1$, in the subsequent analysis, this index ranges from 1 to $K$. When $\tau = K$, the update step from $x_{t,K}$ to $x_{t+1,1}$ is considered a large update step for all $t$. For the other $\tau \in [K-1]$, the subsequent update step is a small step.

We analyze the convergence property of `AdamW-PMA` following the same settings of Kingma & Ba (2014). The metric of interest is the regret, defined as:

$$R_\tau(T) = \sum_{t=1}^{T} f(x_{t,\tau}) - f(x^*), \tag{1}$$

where $\tau$ is the index of the small update steps. We demonstrate that `AdamW-PMA` has an $O(\sqrt{T})$ regret bound, comparable to Adam in the same setting. We use some definitions simplify our notation, where $g_{t,\tau} = \nabla f(x_{t,\tau})$ and $g_{t,\tau,i}$ as the $i$th element.

**Theorem 1.** *Assume that the optimization objective $f$ is convex and has bounded gradients, $\|\nabla f(x)\|_2 \leq G$, $\|\nabla f(x)\|_\infty \leq G_\infty$, and the distance between any parameter generated by `AdamW-PMA` is bounded, $\|x_{t_1,\tau_1} - x_{t_2,\tau_2}\|_2 \leq D$, $\|x_{t_1,\tau_1} - x_{t_2,\tau_2}\|_\infty \leq D_\infty$ for any $t_1, t_2 \in [T]$ and $\tau_1, \tau_2 \in [K]$, and $\beta_1$, $\beta_2$ satisfy $\frac{\sqrt{1-\beta_2}}{1-\beta_1} \leq 1$. `AdamW-PMA` achieves the following regret guarantee, for all $T \geq 1$.*

$$R_K(T) \leq \frac{\sqrt{K}D^2}{2\gamma(1-\beta_1)} \sum_{i=1}^{d} \sqrt{T\hat{v}_{T,K,i}} + \frac{(1+\gamma)K^{\frac{3}{2}}G_\infty}{2(1-\beta_1)} \sum_{i=1}^{d} \|g_{1:KT,i}\|_2 + \frac{D_\infty^2 G_\infty (K-1)}{2(1-\beta_1)}. \tag{2}$$

The proof of Theorem 1 is provided in the Appendix C. Theorem 1 implies that given a horizon $T$, the cumulative regret decreases with the data sparsity, consistent with the theoretical analysis of Adam. Additionally, it is observed that the regret increases with the number of small steps $K$. The intuition behind this relationship is that choosing a larger $K$ can help `AdamW-PMA` converge faster to the optimal point. However, when $K$ is too large, the trajectory of `AdamW-PMA` deviates significantly from the trajectory of Adam, potentially resulting in a large regret.

## 4.2 RESOURCE OVERHEAD ANALYSIS

`AdamW-PMA` does not incur higher memory costs compared to Adam and AdamW with GA. The memory usage of `AdamW-PMA` primarily consists of parameters, gradients, and first and second momenta, which is identical to Adam. When `AdamW-PMA` and Adam with GA use the same small batch size, their memory requirements are equivalent.

## 5 EVALUATION

### 5.1 EXPERIMENTAL SETUP

**Tasks.** Our experiments evaluate `AdamW-PMA` and `Lion-PMA` through language modeling tasks, covering SFT, and DPO (Rafailov et al., 2023) tasks[3]. For the SFT task, we use Phi-2 (Javaheripi et al., 2023) with 2.7B pre-trained parameters and the Alpaca dataset (Taori et al., 2023) as the instruction tuning dataset. For the DPO task, we fine-tune the pre-trained Phi-2 and Qwen1.5-0.5B models on the HH-RLHF-harmless dataset (Bai et al., 2022).

**Baselines.** We mainly compare `AdamW-PMA` and `Lion-PMA` with AdamW and Lion, respectively. For example, after setting a batch size $B$ and a period length $K$ for `AdamW-PMA`[4], we compare it with AdamW with $K$ times of GA[5], whose batch size is $B$, and the equivalent batch size is $KB$ achieved by GA. In each group of experiments, the hyperparameters are the same across all the optimizers. For the `AdamW-PMA` group, we set $lr = 2e-6, 2e-6$ for the two tasks, respectively, and the betas are set to $(0.9, 0.95)$ for `AdamW-PMA` and the baselines in all the experiments. For the `Lion-PMA` group, we set the same learning rate as the `AdamW-PMA` group and let betas be $(0.95, 0.98)$ in all the experiments as in Chen et al. (2023).

**Implementation.** All the following experiments are conducted on a server with $8\times$ NVIDIA A40 GPUs with $8 \times 48G$ GPU memory and Intel(R) Xeon(R) Gold 6330 CPU and Ubuntu 20.04.2. The implementation is based on the Swift framework (Team, 2024b) and PyTorch (Paszke et al., 2019). We set $K = 8, 16$ for the SFT, and DPO tasks, respectively. For the batch size, we set $B = 32, 16$ for each task, respectively.

**Metrics.** For the SFT task, we evaluate the validation loss on the validation dataset and the performance of the trained model on MMLU (Hendrycks et al., 2020) benchmark. For the DPO task, we evaluate the validation loss and the accuracy of classifying the accepted and rejected responses on the validation dataset. Specifically, if the predicted probability of the accepted response is larger than the rejected response, we regard it as a correct classification.

**Methodology of Comparison.** To compare the performance of `AdamW-PMA` and `Lion-PMA` with their baselines, we consider two methodologies of comparing the evaluated metrics. The first methodology is data efficiency. Specifically, we compare the amounts of training data fed into the model when the metrics of the optimizers reach the same level. The intuition behind this comparison methodology is that if an optimizer is faster, it should achieve a specific low loss with fewer training steps. Since the (quasi-equivalent) batch sizes of the optimizers in each group are different, we consider the amount of training data to be fairer to measure the data efficiency instead of the number of steps. The second methodology is comparing the flops. Specifically, we compare the flops of

---

[3]Due to the page limit, some important experiments and results, including pre-training and the impact of learning rate scheduler, are presented in Appendix E.

[4]We abbreviate this setting as `AdamW-PMA`-$K$. So is `Lion-PMA`-$K$.

[5]We abbreviate this setting as AdamW-$K$. So is Lion-$K$.

| Algorithm | Val Loss | MMLU(Zero-Shot) | | | | |
|---|---|---|---|---|---|---|
| | | Hums. | STEM | Social | Other | Avg. |
| AdamW-4 | 0.9212 | 15.4 | 28.3 | 26.7 | 24.3 | 24.4 |
| AdamW-8 | 0.9408 | **19.2** | 22.8 | 26.7 | 25.0 | 23.3 |
| AdamW-PMA-4 | 0.9352 | 16.9 | 22.8 | 25.0 | 22.7 | 21.9 |
| AdamW-PMA-8 | **0.9078** | 16.2 | **28.3** | **30.1** | **35.0** | **27.7** |
| Lion-4 | 0.9227 | 13.1 | 23.3 | 24.2 | 25.7 | 21.8 |
| Lion-8 | 0.9486 | **20.8** | 22.2 | 24.2 | 25.0 | **23.0** |
| Lion-PMA-4 | **0.9136** | 13.1 | **23.3** | **24.2** | 25.7 | 21.8 |
| Lion-PMA-8 | 0.9373 | 17.7 | 22.2 | 22.5 | **26.4** | 22.3 |

Table 1: Comparison of the validation loss and the performance on zero-shot MMLU for various algorithms with $lr = 2e-6$, where validation loss is from the Alpaca dataset after one epoch training. With limited space, we only choose four representative categories and the total average score.

the optimizers when the metrics reach the same level. This intuition is that the flops are the most straightforward metric to measure the speed of an optimizer. If an optimizer is faster, it should achieve a certain level of loss using less time in practice.

## 5.2 SUPERVISED FINE-TUNING (SFT)

Table 1 shows that for AdamW family and Lion family algorithm, our method `PMA` can improve the performance of SFT explicitly. For the validation loss, the `AdamW-PMA` with $K = 8$ (which is what AdamW-PMA-8 refers to. So are the other abbreviations.) is better than other AdamW algorithms, and for Lion family algorithms, `Lion-PMA` with $K = 4$ is better.

In the MMLU-ZS (Zero-Shot) classification tasks, as shown in the table, algorithms incorporating PMA technology achieve superior performance across all categories except for Humanities. Specifically, in the STEM, Social, and Other categories, PMA-enhanced algorithms, consistently outperform their non-PMA counterparts. On average, PMA-enhanced algorithms demonstrate better performance, as indicated by the overall scores (e.g., 27.7 for AdamW-PMA-8).

Interestingly, Table 1 also reveals that algorithms scoring high in Humanities tasks tend to perform poorly in other categories. For instance, AdamW-8 achieves the highest score in Humanities (19.2) within the AdamW group but has one of the lowest overall average scores (23.3). This phenomenon is believed to be caused by the unbalanced SFT data, which lacks sufficient data in Humanities. Conversely, Lion-PMA-4, while maintaining a competitive score in Humanities (17.7), excels in other categories except for the averaged score. The low average score of Lion-PMA-4 is caused by the lack of data in Humanities, lowering the average score despite high scores in many other categories.

## 5.3 DIRECT PREFERENCE OPTIMIZATION (DPO)

**PMA achieves higher accuracy.** We verified the effectiveness of the `PMA`-enhanced optimizers on the DPO task. In Fig. 3a and 3b, we set K to 1 and 16, respectively, and compared the validation accuracy curves of four optimizers from the perspective of total flops. Figure 3c3d use the number of update steps and the number of samples as references, comparing the effects of the four optimizers applied to the DPO task under K=8 and K=16 parameter settings. Among these four optimizers, AdamW is the slowest and achieves the lowest accuracy. When the PMA method is applied to AdamW with smaller values of K = 8, as illustrated in Fig. 3a3c, `AdamW-PMA`'s final convergence accuracy is comparable to that of Lion, which serves as another baseline. Although its convergence speed greatly surpasses that of AdamW, it remains slightly slower than Lion. However, with larger values of K, as shown in Fig. 3b3d, `AdamW-PMA` not only matches Lion in terms of convergence accuracy but also significantly outpaces both AdamW and Lion in terms of convergence speed. Among them, `Lion-PMA` exhibits the best optimization performance. We observed that both `AdamW-PMA` and `Lion-PMA` exhibit significant improvements in both convergence speed and ultimate accuracy compared to AdamW and Lion.

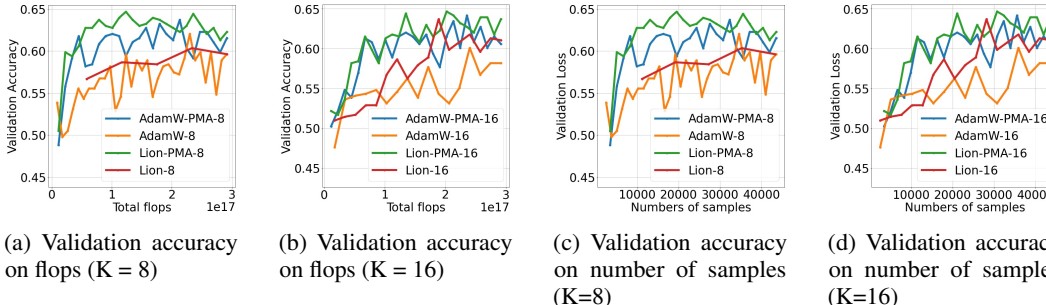

(a) Validation accuracy on flops (K = 8)

(b) Validation accuracy on flops (K = 16)

(c) Validation accuracy on number of samples (K=8)

(d) Validation accuracy on number of samples (K=16)

Figure 3: The accuracy of classifying the accepted and rejected responses on the validation dataset for DPO task. Compared to AdamW and Lion, AdamW-PMA and Lion-PMA exhibit faster convergence rates and higher accuracy.

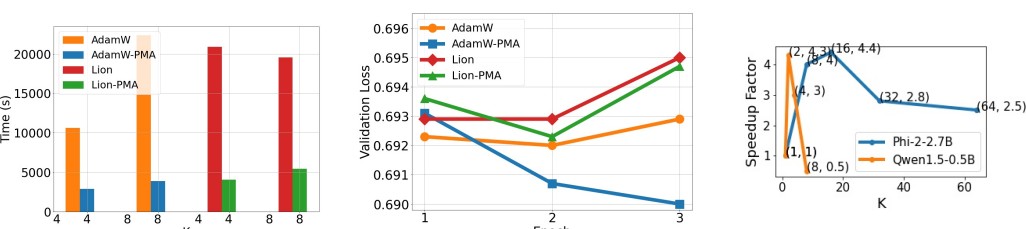

Figure 4: Runtime to achieve the same loss on DPO task. PMA can reduce the training time cost than EMA.

Figure 5: Validation loss of training more epochs on DPO task and Qwen2-0.5B. Lower loss of PMA-based methods demonstrates PMA overfits less.

Figure 6: The speedup factor of AdamW-PMA compared to AdamW under different settings of the hyperparameter K on Phi-2 2.7B and Qwen1.5-0.5B.

**PMA reduces runtime.** We compare the runtime of PMA and EMA-based optimizers to achieve the same validation loss, showing that DPO can reduce the training time. We use the same settings as Fig. 3, and the runtime is illustrated in Fig. 4. On the one hand, results in Fig. 4 show that PMA takes significantly less time to achieve the same validation loss than EMA. On the other hand, after reaching the same loss as EMA, PMA can utilize the left data to achieve a higher accuracy, which aligns with the result in Fig. 3.

**PMA overfits less.** We compare the validation loss of EMA and PMA after more epochs of DPO training. The experiment setting is the same as Fig. 3 but the model is replaced with Qwen2-0.5B. The result is shown in Fig. 5. PMA-based optimizers achieve lower validation losses than EMA-based optimizers, especially after more training epochs. Specifically, the loss of AdamW-PMA achieves a series of decreasing validation loss across the three epochs, compared with the increasing loss of AdamW, showing that PMA can achieve a lower level of over-fitting than EMA.

## 5.4 HYPER-PARAMETER SENSITIVITY

**PMA is sensitive to K.** We conducted experiments using AdamW-PMA and AdamW, setting the hyperparameter $K$ at different values to assess its impact on the speedup factor of PMA. In Fig. 6, we present the results for the DPO task utilizing the Phi-2 and Qwen1.5 model, with AdamW as the baseline. When $K = 1$, AdamW-PMA bypasses the PMA stage and directly reverts to AdamW, thus failing to leverage the variance reduction and acceleration benefits of the PMA method. Conversely, when K is set too high, although the variance in momentum updates is reduced, the excessive reduction in learning rate during the PMA stage leads to a diminished extent of acceleration. When $K = 16$ for Phi-2 and $K = 2$ for Qwen, the reduction in the variance of the momentum updates and the decrease in learning rate achieve a relatively optimal balance, therefore the application of the PMA method achieved the highest observed speedup.

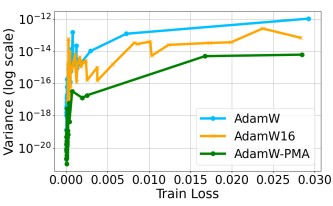 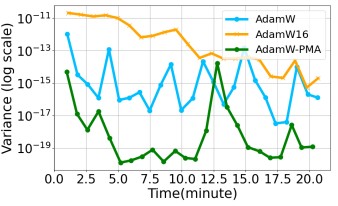 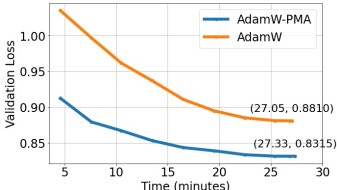

(a) Variance vs Training loss          (b) Variance vs Time

Figure 7: Comparison of the magnitude in variance with respect to the training loss and time for our algorithm versus AdamW. The vertical coordinates all use log scale since our algorithm is orders of magnitude different from other algorithms.

Figure 8: Validation loss of SFT on Llama2-7B. AdamW-PMA takes a similar time as AdamW, but achieves a much lower loss.

### 5.5 OTHER PROPERTIES

**PMA reduces variance.** We demonstrate that PMA can achieve a lower variance of the update direction than that of EMA. For our experiments, we employ the GPT-2 medium (Brown et al., 2020) model with 350M parameters, and utilize the Alpaca dataset with all three algorithms configured identically: $lr = 5e-6$, and betas are set to (0.9, 0.95). The GA step and K value in Alg. 1 are both set to 16. We use the last layer gradient to approximate the gradient of the whole model (Ash et al., 2019; Mirzasoleiman et al., 2020; Killamsetty et al., 2021b;a).

Figure 7a shows that, at equivalent levels of training loss, our algorithm exhibits lower gradient update variance. Furthermore, as depicted in Fig. 7b, the update variance of our algorithm consistently remains substantially lower than that of the benchmark throughout the training duration.

**PMA can scale up and be quantized.** We evaluate the performance of `AdamW-PMA` on a 7B-level BF16 model, to demonstrate that `PMA` can scale up on larger models. The experiment is conducted on Llama2-7B-base quantized to BF16 and SFT on the DuReader_Robust dataset (Tang et al., 2020). The model is trained for one epoch. The statistics of the validation loss are plotted in Fig. 8. `AdamW-PMA` achieves lower validation loss than AdamW across the whole training process, demonstrating the superiority of `PMA` than EMA.

**PMA costs a little more time.** As shown in Fig. 8, `AdamW-PMA` takes about 2% more time than Adam when training a 7B model, indicating that although there are more update steps and more communication overhead in `AdamW-PMA`, these small update steps do not take too much time.

## 6 DISCUSSION AND CONCLUSION

We address the problem of high-variance stochastic optimization on GPU-memory-limited devices for training LLMs. We identified that the low convergence rate of current momentum-based optimizers is primarily due to the EMA method, which forgets historical gradients too quickly, thus failing to leverage them effectively for stabilizing updates. To tackle this, we propose `PMA`, a new momentum update method that splits the training process into periods and applies a vanilla moving average within each period. This approach assigns a higher weight to historical gradients, thereby stabilizing updates when gradient variance is high. We modify AdamW and Lion using `PMA`, resulting in `AdamW-PMA` and `Lion-PMA`, respectively. Empirical evaluations on SFT and DPO tasks using the Phi-2 and Qwen model demonstrate that `PMA` achieves approximately $2\times$ speedup in the training process and delivers better performance on downstream tasks.

However, `PMA` modified methods could incur higher communication overhead in multi-GPU training scenarios, especially when $K$ is large. For example, Since `AdamW-PMA` employs extra steps of parameter update during GA, more communication overhead is required when multiple GPUs are employed for the training task. Specifically, since there are $K$ more communication rounds in `AdamW-PMA` than in Adam with GA, the communication cost of `AdamW-PMA` is $K$ times higher than that of Adam with GA.

## REPRODUCIBILITY STATEMENT

The code of the experiment is attached in the supplementary material as a zip file. Please refer to the `README_ICLR_submission.md` for detailed usage. The proof is provided in the Appendix.

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

# Appendix

## Table of Contents

## A  RELATED WORK

**First-Order Adaptive Methods.**  The basic idea behind designing adaptive first-order optimizers is to adapt the direction and learning rate for each parameter individually. AdaGrad (Duchi et al., 2011) achieves this by adjusting the learning rate of features based on estimated geometry and assigning larger learning rates to infrequent features. RMSProp (Hinton et al., 2012) enhances AdaGrad by introducing a running average of the second-order momentum, i.e., the square of the gradients. Adam (Kingma & Ba, 2014) further improves RMSProp by introducing a running average of gradients. Alongside its enhanced version with weight decay, AdamW (Loshchilov & Hutter, 2017), Adam has emerged as the predominant approach for solving optimization problems in deep learning, particularly in training Transformers (Vaswani et al., 2017). Numerous subsequent works can be viewed as variants of first-order adaptive methods (Dozat, 2016; Shazeer & Stern, 2018; Reddi et al., 2019; Zhuang et al., 2020; You et al., 2019; Xie et al., 2022; Chen et al., 2023). The main drawback of Adam-like methods is its memory cost. The optimizers maintains the first and second momentum along with the current gradient, leading to a heavy load for memory-constrained devices.

**Memory-Efficient Optimizers.**  Adafactor (Shazeer & Stern, 2018) reduces the memory by only maintaining the row and column sum of the second order momentum and estimate the second moments based on these sums. LOMO (Lv et al., 2023) fuses the gradient computation and the parameter

update in one step to reduce memory usage. CAME (Luo et al., 2023) supports adaptive confidence-based updating guided by the residual between predicted update and generated update. GaLore (Zhao et al., 2024) uses a low-rank projected gradient to save memory and full parameter update to achieve a good performance. Adam-mini (Zhang et al., 2024) reduces memory by cutting down the learning rate resources in Adam. However, it is hardly practical to deploy the above mentioned methods to memory-constrained devices. The reason includes two folds. On the one hand, all these methods, except Adam-mini, suffer from lower convergence rates than Adam, meaning that they are saving memory in the cost of speed. On the other hand, their saved memory is not enough for memory-constrained devices, especially when training LLMs. For example, CAME can save $12.1\%$ of Adam's memory cost (according to Table 1 of Luo et al. (2023)), which is far from enough if one is going to use a large batch size and a scaled-up model. Considering the impracticality of these memory-efficient optimizers, we shall state the importance of applying GA and the necessity of accelerating GA.

**Variance Reduction.** The variance reduction techniques in SGD (Bottou et al., 2018) include dynamic sampling, gradient aggregation, and iterate averaging. As for optimizer design, we focus on the last two techniques. The gradient aggregation methods reduce variance by reusing previously computed information. Specifically, at time step $t$, SVRG (Johnson & Zhang, 2013) maintains a copy of the historical parameter $\theta_k$ where $k < t$. It computes a batched gradient $G_{\theta_k} = \frac{1}{n}\sum_{i=1}^{n}\nabla f_{\theta_k}(x_i)$ and derives an unbiased estimator of the current gradient by $\mathbb{E}[\nabla R_{\theta_t}] = \nabla f_{\theta_t}(x_t) - (\nabla f_{\theta_k}(x_t) - G_{\theta_k})$, where $x_t$ is a sample from the input space. SAGA (Defazio et al., 2014) stores the historical gradient for each data sample and estimates the current gradient using the average of the historical gradients. The iterate averaging method (Polyak, 1991) stores the parameters after each SGD step and returns the average of the stored parameters. Nesterov (2013) employs gradient aggregation and yield $O(1/t)$ rate of convergence for the averaged iterate sequence.

# B   ADDITIONAL THEORETICAL ANALYSIS

In this section, we provide a theoretical analysis on the convergence property of `AdamW-PMA`. Specifically, we focus on the convergence properties concerning the number of large update steps. This focus is due to the time cost between two large steps being approximately equal to the time between two updates of Adam with GA. During the analysis, we slightly modify the notations for ease of analysis. Unlike Algorithm 1, where the index of small update steps ranges from 0 to $K - 1$, in the subsequent analysis, this index ranges from 1 to $K$. Specifically, when $\tau = K$, the update step from $x_{t,K}$ to $x_{t+1,1}$ is considered a large update step for all $t$. For the other $\tau \in [K-1]$, the subsequent update step is a small step.

Firstly, we can show the average regret of `AdamW-PMA` converges based on Theorem 1,

**Corollary 1.** *Assume that the optimization objective $f$ is convex and has bounded gradients, $\|\nabla f(x)\|_2 \leq G$, $\|\nabla f(x)\|_\infty \leq G_\infty$, and the distance between any parameter generated by `AdamW-PMA` is bounded, $\|x_{t_1,\tau_1} - x_{t_2,\tau_2}\|_2 \leq D$, $\|x_{t_1,\tau_1} - x_{t_2,\tau_2}\|_\infty \leq D_\infty$ for any $t_1, t_2 \in [T]$ and $\tau_1, \tau_2 \in [K]$, and $\beta_1$, $\beta_2$ satisfy $\frac{\sqrt{1-\beta_2}}{1-\beta_1} \leq 1$. `AdamW-PMA` achieves the following regret guarantee, for all $T \geq 1$.*

$$\frac{R_K(T)}{T} = O\left(\frac{1}{\sqrt{T}}\right).$$

Then, we provide the update size between two large update steps in general non-convex settings.

**Theorem 2.** *Assume that the objective function $f$ is $L$-smooth, the step size between two large update steps is bounded by*

$$\|x_{t+1,1} - x_{t,1}\|^2 \leq \frac{2}{L}\left(1 + \frac{\gamma^2 L}{\sqrt{K}}\frac{(1-\beta_1)^2}{1-\beta_2}(K+1)\right) \cdot \bar{\zeta}(2a)^{2t-2}, \tag{3}$$

*where $\bar{\zeta}$ and $a$ are constants, and $\bar{\zeta}(2a)^{2t-2} \geq \zeta(2a)^{2t-2} + \frac{c}{1-4a^2} + K^2$, and $a = \frac{\beta_1(1-\beta_1)}{\sqrt{\beta_2(1-\beta_2)}} \cdot \frac{1}{\sqrt{K}}$.*

Theorem 2 indicates that the distance between two large update steps is bounded and converges to 0. Despite having $K$ small updates with varying momentum averaging weights, the step sizes still converge rapidly , suggesting the validity of setting the learning rate of the small steps to be $\gamma/\sqrt{K}$. Furthermore, the exponential term decreases with $K$, aligning with the intuition that more small update steps lead to faster convergence.

## C   PROOF OF THEOREM 2

Before the analysis, we slightly modify the notations to simplify the analysis. The large step update takes the $x_{t,K}$ as input and outputs $x_{t+1,1}$. Then, `AdamW-PMA` uses small step update to obtain $x_{t+1,2}, \ldots, x_{t+1,K}$. It is noteworthy that the indexes of small step updates in Algorithm 1 range from $0$ to $K-1$, while in the following analysis, they will range from $1$ to $K$.

Before the analysis, we start with some important lemmas. Firstly, we consider the size of small step updates between two large step updates. To start with, we bound the size of every small step update using Lemma 1.

**Lemma 1.** *When $\tau \geq 2$,*

$$\left\| \frac{m_{t,\tau}}{\sqrt{v_{t,\tau}}} \right\| \leq \frac{1 - \beta_1}{\sqrt{1 - \beta_2}} \cdot \frac{1}{\sqrt{\tau}} \left( \left\| \frac{m_{t,1}}{\sqrt{v_{t,1}}} \right\| + \tau - 1 \right). \tag{4}$$

*Proof.*

$$
\begin{aligned}
\left\| \frac{m_{t,\tau}}{\sqrt{v_{t,\tau}}} \right\| &= \left\| \frac{\frac{\tau-1}{\tau} m_{t,\tau-1} + \frac{1-\beta_1}{\tau} g_{t,\tau-1}}{\sqrt{\frac{\tau-1}{\tau} v_{t,\tau-1} + \frac{1-\beta_2}{\tau} g_{t,\tau-1}^2}} \right\| \\
&\leq \frac{1 - \beta_1}{\sqrt{1 - \beta_2}} \cdot \frac{1}{\sqrt{\tau}} \cdot \left\| \frac{m_{t,1} + \sum_{\sigma=2}^{\tau} g_{t,\sigma}}{\sqrt{v_{t,1} + \sum_{\sigma=2}^{\tau} g_{t,\sigma}^2}} \right\| \\
&\leq \frac{1 - \beta_1}{\sqrt{1 - \beta_2}} \cdot \frac{1}{\sqrt{\tau}} \left( \sum_{\sigma=2}^{\tau} \left\| \frac{g_{t,\sigma}}{\sqrt{v_{t,1} + \sum_{\rho=2}^{\tau} g_{t,\rho}^2}} \right\| + \left\| \frac{m_{t,1}}{\sqrt{v_{t,1} + \sum_{\sigma=2}^{\tau} g_{t,\sigma}^2}} \right\| \right) \\
&\leq \frac{1 - \beta_1}{\sqrt{1 - \beta_2}} \cdot \frac{1}{\sqrt{\tau}} \left( \left\| \frac{m_{t,1}}{\sqrt{v_{t,1}}} \right\| + \sum_{\sigma=2}^{\tau} \left\| \frac{g_{t,\sigma}}{\sqrt{g_{t,\sigma}^2}} \right\| \right) \\
&= \frac{1 - \beta_1}{\sqrt{1 - \beta_2}} \cdot \frac{1}{\sqrt{\tau}} \left( \left\| \frac{m_{t,1}}{\sqrt{v_{t,1}}} \right\| + \tau - 1 \right)
\end{aligned}
$$

$\square$

Then, we bound the squared size of the small step update.

**Corollary 2.**

$$\left\| \frac{m_{t,\tau}}{\sqrt{v_{t,\tau}}} \right\|^2 \leq \frac{(1 - \beta_1)^2}{1 - \beta_2} \cdot \frac{2}{\tau} \left( \left\| \frac{m_{t,1}}{\sqrt{v_{t,1}}} \right\|^2 + \tau^2 \right). \tag{5}$$

*Proof.* Since $(a + b)^2 = a^2 + b^2 + 2ab \leq 2(a^2 + b^2)$,

$$
\begin{aligned}
\left\| \frac{m_{t,\tau}}{\sqrt{v_{t,\tau}}} \right\| &\leq \frac{1 - \beta_1}{\sqrt{1 - \beta_2}} \cdot \frac{1}{\sqrt{\tau}} \left( \left\| \frac{m_{t,1}}{\sqrt{v_{t,1}}} \right\| + \tau - 1 \right) \\
&\leq \frac{(1 - \beta_1)^2}{1 - \beta_2} \cdot \frac{2}{\tau} \left( \left\| \frac{m_{t,1}}{\sqrt{v_{t,1}}} \right\|^2 + \tau^2 \right).
\end{aligned}
$$

$\square$

Then, we bound the sum of the squared size of small step updates between two large step updates.

**Corollary 3.**

$$\sum_{\sigma=1}^{\tau-1} \left\| \frac{m_{t,\sigma}}{\sqrt{v_{t,\sigma}}} \right\|^2 \leq \frac{(1 - \beta_1)^2}{1 - \beta_2} \sum_{\sigma=1}^{\tau-2} \frac{2}{\sigma} \left( \left\| \frac{m_{t,1}}{\sqrt{v_{t,1}}} \right\|^2 + \sigma^2 \right) + \left\| \frac{m_{t,1}}{\sqrt{v_{t,1}}} \right\|^2. \tag{6}$$

*Proof.*

$$\sum_{\sigma=1}^{\tau-1} \left\| \frac{m_{t,\sigma}}{\sqrt{v_{t,\sigma}}} \right\|^2 = \sum_{\sigma=1}^{\tau-2} \left\| \frac{m_{t,\sigma}}{\sqrt{v_{t,\sigma}}} \right\|^2 + \left\| \frac{m_{t,\tau-1}}{\sqrt{v_{t,\tau-1}}} \right\|^2$$

$$\underset{\text{Corollary } 2}{\leq} \sum_{\sigma=1}^{\tau-2} \left\| \frac{m_{t,\sigma}}{\sqrt{v_{t,\sigma}}} \right\|^2 + \frac{(1-\beta_1)^2}{1-\beta_2} \cdot \frac{2}{\tau-1} \cdot \left( \left\| \frac{m_{t_1}}{\sqrt{v_{t,1}}} \right\|^2 + (\tau-1)^2 \right)$$

$$\leq \frac{(1-\beta_1)^2}{1-\beta_2} \sum_{\sigma=1}^{\tau-2} \frac{2}{\sigma} \left( \left\| \frac{m_{t,1}}{\sqrt{v_{t,1}}} \right\|^2 + \sigma^2 \right) + \left\| \frac{m_{t,1}}{\sqrt{v_{t,1}}} \right\|^2.$$

$\square$

After bounding the small steps between two large update steps, we consider the size of a large update step and $K$ following small steps. To start with, we assume that the update size of the first large step is bounded.

**Assumption 1.** *Let $a = \frac{\beta_1(1-\beta_1)}{\sqrt{\beta_2(1-\beta_2)}} \cdot \frac{1}{\sqrt{K}}$ and $b = \frac{1-\beta_1}{\sqrt{1-\beta_2}} \cdot \frac{1}{\sqrt{K}} \cdot \left(1 + \frac{\beta_1 K}{\sqrt{\beta_2}}\right)$.*

$$\left\| \frac{m_{1,1}}{\sqrt{v_{1,1}}} \right\| \leq \frac{b}{1-a} + \alpha. \tag{7}$$

Then, we make some assumptions on the weight of the momentum.

**Assumption 2.** *For all $t$,*

$$\frac{\sqrt{1-\beta_2^t}}{1-\beta_1^t} \leq 1.$$

If we take $\beta_1 = 0.9$, $\beta_2 = 0.99$ as the default configuration of Adam, this assumption holds.

Then, we bound the size of a large update step.

**Lemma 2.** *By tuning the hyper-parameters $\beta_1$ and $\beta_2$, let $a \leq 1/2$. Then*

$$\left\| \frac{m_{t,1}}{\sqrt{v_{t,1}}} \right\| \leq \bar{\alpha} \cdot a^{t-1}, \tag{8}$$

*where $\bar{\alpha} > \alpha$ is a constant to make $\bar{\alpha} \cdot a^{t-1} \geq \alpha \cdot a^{t-1} + \frac{b}{1-a} + K$.*

*Proof.*

$$\left\| \frac{m_{t,1}}{\sqrt{v_{t,1}}} \right\| = \left\| \frac{\beta_1 m_{t-1,K} + \frac{1-\beta_1}{K} g_{t,1}}{\sqrt{\beta_2 v_{t-1,K} + \frac{1-\beta_2}{K} g_{t,1}^2}} \right\|$$

$$\leq \left\| \frac{\beta_1 m_{t-1,K}}{\sqrt{\beta_2 v_{t-1,K}}} \right\| + \frac{1-\beta_1}{\sqrt{1-\beta_2}} \cdot \frac{1}{\sqrt{K}} \cdot \left\| \frac{g_{t,1}}{\sqrt{g_{t,1}^2}} \right\|$$

$$= \left\| \frac{\beta_1 m_{t-1,K}}{\sqrt{\beta_2 v_{t-1,K}}} \right\| + \frac{1-\beta_1}{\sqrt{1-\beta_2}} \cdot \frac{1}{\sqrt{K}}$$

$$\underset{\text{Lem. } 1}{\leq} \frac{1-\beta_1}{\sqrt{1-\beta_2}} \cdot \frac{1}{\sqrt{K}} + \frac{\beta_1}{\sqrt{\beta_2}} \cdot \frac{1-\beta_1}{\sqrt{1-\beta_2}} \cdot \frac{1}{\sqrt{K}} \cdot \left( \left\| \frac{m_{t-1,1}}{\sqrt{v_{t-1,1}}} \right\| + K - 1 \right)$$

$$\leq \underbrace{\frac{\beta_1(1-\beta_1)}{\sqrt{\beta_2(1-\beta_2)}} \cdot \frac{1}{\sqrt{K}}}_{:=a} \cdot \left\| \frac{m_{t-1,1}}{\sqrt{v_{t-1,1}}} \right\| + \underbrace{\frac{1-\beta_1}{\sqrt{1-\beta_2}} \cdot \frac{1}{\sqrt{K}} \cdot \left(1 + \frac{\beta_1 K}{\sqrt{\beta_2}}\right)}_{:=b}$$

Let $x_t = \left\| \frac{m_{t,1}}{\sqrt{v_{t,1}}} \right\|$, then

$$x_t \leq a^{t-1} \left( x_1 - \frac{b}{1-a} \right) + \frac{b}{1-a}$$

Then, by Assumption 1

$$\left\| \frac{m_{t,1}}{\sqrt{v_{t,\tau}}} \right\| \leq \alpha \cdot a^{t-1} + \frac{b}{1-a} \leq \bar{\alpha} \cdot a^{t-1}.$$

□

Similar to the above approach, we assume the bounded squared first large step and prove the bounded squared large steps.

**Assumption 3.** *Let* $c = \frac{(1-\beta_1)^2}{1-\beta_2} \cdot \frac{2}{K} \left( 1 + \frac{2\beta_1^2 K^2}{\beta_2^2} \right)$

$$\left\| \frac{m_{1,1}}{\sqrt{v_{t,1}}} \right\|^2 \leq \frac{c}{1-4a^2} + \zeta \tag{9}$$

**Corollary 4.**

$$\left\| \frac{m_{t,1}}{\sqrt{v_{t,1}}} \right\|^2 \leq \bar{\zeta}(2a)^{2t-2}, \tag{10}$$

*where* $\bar{\zeta} > \zeta$ *is a constant to make* $\bar{\zeta}(2a)^{2t-2} \geq \zeta(2a)^{2t-2} + \frac{c}{1-4a^2} + K^2$.

*Proof.*

$$\left\| \frac{m_{t,1}}{\sqrt{v_{t,1}}} \right\|^2 \leq 2 \left\| \frac{\beta_1 m_{t-1,K}}{\sqrt{\beta_2 v_{t-1,K}}} \right\|^2 + \frac{(1-\beta_1)^2}{1-\beta_2} \cdot \frac{2}{K}$$

$$\leq \frac{(1-\beta_1)^2}{1-\beta_2} \cdot \frac{2}{K} + \frac{4}{K} \cdot \frac{(1-\beta_1)^2 \beta_1^2}{(1-\beta_2)\beta_2} \cdot \left( \left\| \frac{m_{t-1,1}}{\sqrt{v_{t-1,1}}} \right\|^2 + K^2 \right)$$

$$= 4a^2 \left\| \frac{m_{t-1,1}}{\sqrt{v_{t-1,1}}} \right\|^2 + c.$$

Then,

$$\left\| \frac{m_{t,1}}{\sqrt{v_{t,1}}} \right\|^2 \leq (2a)^{2t-2} \left( \left\| \frac{m_{1,1}}{\sqrt{v_{1,1}}} \right\|^2 - \frac{c}{1-4a^2} \right) + \frac{c}{1-4a^2}$$

$$\leq \zeta(2a)^{2t-2} + \frac{c}{1-4a^2}$$

$$\leq \bar{\zeta}(2a)^{2t-2}.$$

□

Before proving Theorem 2, we need more assumptions on the objective function and the initial point. First, we assume that $f$ has Lipschitz continuous gradient.

**Assumption 4** (*L*-smoothness)**.** *A function* $f: \mathbb{R}^d \to \mathbb{R}$ *is differentiable and for any* $x_1, x_2 \in \mathbb{R}^d$,

$$\|\nabla f(x_1) - \nabla f(x_2)\| \leq L\|x_1 - x_2\|,$$

*where* $L$ *is a constant.*

Now, by putting everything together, we are ready to prove Theorem 2.

*Proof of Theorem 2.* Let $\mathcal{T} = \frac{L}{2}\|x_{t+1,1} - x_t\|^2$. At the beginning, we assume that the AdamW-PMA does not employ the bias correction shown in Line 12-13 in Algorithm 1.

$$\mathcal{T} = \frac{L}{2}\|x_{t+1,1} - x_{t.1}\|^2$$

$$\leq \frac{L}{2}\sum_{\tau=1}^{K-1}\|x_{t,\tau+1} - x_{t,\tau}\|^2 + \frac{L}{2}\|x_{t+1,1} - x_{t,K}\|^2$$

$$= \frac{L}{2}\sum_{\tau=1}^{K-1}\left\|\frac{\gamma}{\sqrt{K}}\frac{\hat{m}_{t,\tau}}{\sqrt{\hat{v}_{t,\tau}}}\right\|^2 + \frac{L}{2}\left\|\gamma\cdot\frac{\hat{m}_{t,K}}{\sqrt{\hat{v}_{t,K}}}\right\|^2$$

$$= \frac{\gamma^2 L}{2\sqrt{K}}\sum_{\tau=1}^{K-1}\left\|\frac{m_{t,\tau}}{\sqrt{v_{t,\tau}}}\right\|^2 + \frac{\gamma^2 L\sqrt{K}}{2}\left\|\frac{m_{t,K}}{\sqrt{v_{t,K}}}\right\|^2.$$

$$\mathcal{T} \leq \frac{\gamma^2 L}{2\sqrt{K}}\sum_{\tau=1}^{K-1}\left\|\frac{m_{t,\tau}}{\sqrt{v_{t,\tau}}}\right\|^2 + \frac{\gamma^2 L\sqrt{K}}{2}\left\|\frac{m_{t,K}}{\sqrt{v_{t,K}}}\right\|^2$$

$$\underset{\leq}{\text{Corollary 2}} \quad \frac{\gamma^2 L}{2\sqrt{K}}\sum_{\tau=1}^{K-1}\left\|\frac{m_{t,\tau}}{\sqrt{v_{t,\tau}}}\right\|^2 + \frac{\gamma^2 L}{2}\frac{(1-\beta_1)^2}{1-\beta_2}\cdot\frac{2}{\sqrt{K}}\left(\left\|\frac{m_{t,1}}{\sqrt{v_{t,1}}}\right\|^2 + K^2\right)$$

$$= \frac{\gamma^2 L}{2\sqrt{K}}\sum_{\tau=1}^{K-1}\left\|\frac{m_{t,\tau}}{\sqrt{v_{t,\tau}}}\right\|^2 + \frac{\gamma^2 L}{\sqrt{K}}\cdot\frac{(1-\beta_1)^2}{1-\beta_2}\cdot\left(\left\|\frac{m_{t,1}}{\sqrt{v_{t,1}}}\right\|^2 + K^2\right)$$

$$\underset{\leq}{\text{Corollary 3}} \quad \frac{\gamma^2 L}{2\sqrt{K}}\frac{(1-\beta_1)^2}{1-\beta_2}\sum_{\sigma=1}^{K-2}\frac{2}{\sigma}\left(\left\|\frac{m_{t,1}}{\sqrt{v_{t,1}}}\right\|^2 + \sigma^2\right) + \left\|\frac{m_{t,1}}{\sqrt{v_{t,1}}}\right\|^2 + \frac{\gamma^2 L}{\sqrt{K}}\cdot\frac{(1-\beta_1)^2}{1-\beta_2}\cdot\left(\left\|\frac{m_{t,1}}{\sqrt{v_{t,1}}}\right\|^2 + K^2\right)$$

$$\leq \frac{\gamma^2 L}{\sqrt{K}}\frac{(1-\beta_1)^2}{1-\beta_2}\left(K\cdot\left\|\frac{m_{t,1}}{\sqrt{v_{t,1}}}\right\|^2 + K^2\right) + \left(1 + \frac{\gamma^2 L}{\sqrt{K}}\frac{(1-\beta_1)^2}{1-\beta_2}\right)\left\|\frac{m_{t,1}}{\sqrt{v_{t,1}}}\right\|^2 + \gamma^2 LK^{\frac{3}{2}}\frac{(1-\beta_1)^2}{1-\beta_2}$$

$$= \left(1 + \frac{\gamma^2 L}{\sqrt{K}}\frac{(1-\beta_1)^2}{1-\beta_2}(K+1)\right)\cdot\left\|\frac{m_{t,1}}{\sqrt{v_{t,1}}}\right\|^2 + 2\gamma^2 LK^{\frac{3}{2}}\frac{(1-\beta_1)^2}{1-\beta_2}$$

$$\underset{\leq}{\text{Corollary 4}} \quad \left(1 + \frac{\gamma^2 L}{\sqrt{K}}\frac{(1-\beta_1)^2}{1-\beta_2}(K+1)\right)\cdot\bar{\zeta}(2a)^{2t-2} + 2\gamma^2 LK^{\frac{3}{2}}\frac{(1-\beta_1)^2}{1-\beta_2}$$

$$\underset{\leq}{\text{Larger }\bar{\zeta}} \quad \left(1 + \frac{\gamma^2 L}{\sqrt{K}}\frac{(1-\beta_1)^2}{1-\beta_2}(K+1)\right)\cdot\bar{\zeta}(2a)^{2t-2}.$$

Thus,

$$\|x_{t+1,1} - x_{t,1}\|^2 \leq \frac{2}{L}\left(1 + \frac{\gamma^2 L}{\sqrt{K}}\frac{(1-\beta_1)^2}{1-\beta_2}(K+1)\right)\cdot\bar{\zeta}(2a)^{2t-2}.$$

Then, we consider the bias correction shown in Linw 12-13 in Algorithm 1. By the bias correction, the learning rate at time step $t$ can be viewed as $\gamma_t = \frac{\sqrt{1-\beta_2^t}}{1-\beta_1}\gamma \leq \gamma$ by Assumption 2. Then, with the bias correction operation, this bound still holds.

$\square$

# D PROOF OF THEOREM 1

Before the analysis, we assume that the variable is bounded, as assumed in Kingma & Ba (2014).

**Assumption 5.** *We assume that the distance between the variable and the optimal point is bounded during the optimization process, such that $\|x_{t,\tau} - x^*\|_2 \leq D$, $\|x_{i,j} - x_{k,l}\|_\infty \leq D_\infty$.*

*Proof of Theorem 1.* Since the objective function $f$ is convex,

$$f(x_{t,K}) - f(x^*) \leq \langle\nabla f(x_{t,K}), x_{t,K} - \theta^*\rangle = \sum_{i=1}^{d}g_{t,K,i}(x_{t,K,i} - x_i^*).$$

Using the update method defined in Algorithm 1, we can get

$$x_{t+1,1,i} = x_{t,K,i} - \gamma \frac{\hat{m}_{t,K,i}}{\sqrt{\hat{v}_{t,K,i}}}$$

$$= x_{t,K,i} - \frac{\gamma}{1-\beta_1^t} \left( \frac{K-1}{\sqrt{K}\sqrt{v_{t,K,i}}} m_{t,K-1,i} + \frac{1-\beta_1}{\sqrt{K}\sqrt{v_{t,K,i}}} g_{t,K,i} \right).$$

$$(x_{t+1,1,i} - x^*)^2 = (x_{t,K,i} - x_i^*)^2 - \frac{2\gamma}{1-\beta_1^t}(x_{t,K,i} - x_i^*)\left( \frac{K-1}{\sqrt{K}\sqrt{v_{t,K,i}}} m_{t,K-1,i} + \frac{1-\beta_1}{\sqrt{K}\sqrt{v_{t,K,i}}} g_{t,K,i} \right)$$

$$+ \gamma^2 K \left( \frac{m_{t,K,i}}{\sqrt{v_{t,K,i}}} \right)^2.$$

Rearrange the equation above,

$$g_{t,K,i}(x_{t,K,i} - x_i^*) = \frac{(1-\beta_1^t)\sqrt{K}\sqrt{v_{t,K,i}}}{2\gamma(1-\beta_1)}\left( (x_{t+1,K,i} - x_i^*)^2 - (x_{t,K,i} - x_i^*)^2 \right)$$

$$+ \frac{K-1}{1-\beta_1} m_{t,K-1,i}(x_{t,K,i} - x_i^*) + \frac{(1-\beta_1^t)\gamma K^{\frac{3}{2}}\sqrt{v_{t,K,i}}}{2(1-\beta_1)}\left( \frac{m_{t,K,i}}{\sqrt{v_{t,K,i}}} \right)^2$$

$$\leq \frac{\sqrt{K}\sqrt{v_{t,K,i}}}{2\gamma(1-\beta_1)}\left( (x_{t+1,K,i} - x_i^*)^2 - (x_{t,K,i} - x_i^*)^2 \right)$$

$$+ \frac{K-1}{1-\beta_1}(x_{t,K,i} - x_i^*)\sqrt{v_{t,K-1}}\left( \frac{m_{t,K-1,i}}{\sqrt{v_{t,K-1,i}}} \right) + \frac{\gamma K^{\frac{3}{2}}}{2(1-\beta_1)}\frac{m_{t,K,i}^2}{\sqrt{v_{t,K,i}}}$$

$$\leq \frac{\sqrt{K}\sqrt{v_{t,K,i}}}{2\gamma(1-\beta_1)}\left( (x_{t+1,K,i} - x_i^*)^2 - (x_{t,K,i} - x_i^*)^2 \right)$$

$$+ \frac{K-1}{2(1-\beta_1)}(x_{t,K,i} - x^*)^2 \cdot \sqrt{v_{t,K,i}} + \frac{K-1}{2(1-\beta_1)}\frac{m_{t,K-1,i}^2}{\sqrt{v_{t,K-1,i}}} + \frac{\gamma K^{\frac{3}{2}}}{2(1-\beta_1)}\frac{m_{t,K,i}^2}{\sqrt{v_{t,K,i}}}.$$

$$R_K(T) \leq \sum_{t=1}^{T}\sum_{i=1}^{d} g_{t,K,i}(x_{t,K,i} - x_i^*)$$

$$\leq \sum_{i=1}^{d}\sum_{t=1}^{T} \frac{\sqrt{K}\sqrt{v_{t,K,i}}}{2\gamma(1-\beta_1)}(x_{t+1,K,i} - x_i^*)^2 - \frac{\sqrt{K}\sqrt{v_{t,K,i}}}{2\gamma(1-\beta_1)}(x_{t,K,i} - x_i^*)^2$$

$$+ \frac{K-1}{2(1-\beta_1)}(x_{t,K,i} - x^*)^2 \cdot \sqrt{v_{t,K,i}} + \frac{K-1}{2(1-\beta_1)}\frac{m_{t,K-1,i}^2}{\sqrt{v_{t,K-1,i}}} + \frac{\gamma K^{\frac{3}{2}}}{2(1-\beta_1)}\frac{m_{t,K,i}^2}{\sqrt{v_{t,K,i}}}$$

$$\overset{\text{Lemma 2}}{\leq} \frac{\sqrt{K}D^2}{2\gamma(1-\beta_1)}\sum_{i=1}^{d}\sqrt{T\hat{v}_{T,K,i}} + \frac{D_\infty^2(K-1)}{2(1-\beta_1)} \cdot \sum_{i=1}^{d}\sum_{t=1}^{T}\sqrt{v_{T,K,i}}$$

$$+ \frac{(1+\gamma)K^{\frac{3}{2}}G_\infty}{2(1-\beta_1)}\sum_{i=1}^{d}\|g_{1:KT,i}\|_2$$

$$\leq \frac{\sqrt{K}D^2}{2\gamma(1-\beta_1)}\sum_{i=1}^{d}\sqrt{T\hat{v}_{T,K,i}} + \frac{(1+\gamma)K^{\frac{3}{2}}G_\infty}{2(1-\beta_1)}\sum_{i=1}^{d}\|g_{1:KT,i}\|_2 + \frac{D_\infty^2 G_\infty(K-1)}{2(1-\beta_1)}.$$

$$\square$$

# E ADDITIONAL EXPERIMENTS

## E.1 PRE-TRAINING

Although `PMA` is designed for post-training, we also evaluate its performance on pre-training task. Specifically, we train a randomly-initialized nanoGPT model on WikiPedia dataset. Figure 9 shows

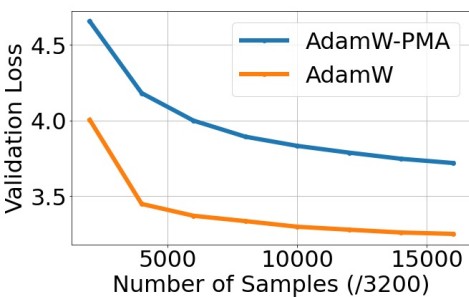

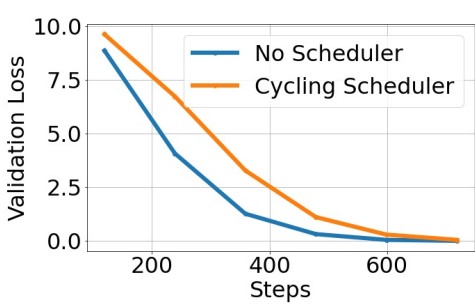

Figure 9: Runtime to achieve the same loss on DPO task. PMA can reduce the training time cost than EMA.

Figure 10: Validation loss of AdamW without learning rate scheduler and AdamW with a PMA-like lr scheduler.

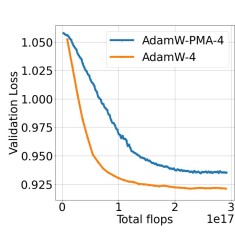
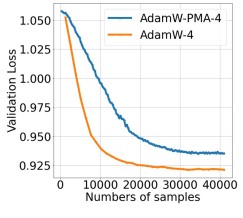
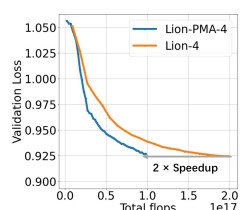
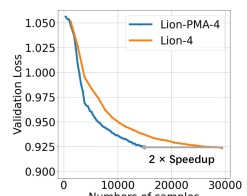

(a) AdamW-PMA v.s. AdamW on flops

(b) AdamW-PMA v.s. AdamW on number of samples

(c) Lion-PMA v.s. Lion on flops

(d) Lion-PMA v.s. Lion on number of samples

Figure 11: From the perspectives of total flops and number of steps, AdamW-PMA and Lion-PMA achieved speedups of 1.8x and 1.4x respectively, compared to AdamW and Lion when K = 1.

the validation loss of `AdamW-PMA` and AdamW. EMA-based AdamW achieves a lower validation loss than `AdamW-PMA`. This is because `PMA`, especially the small update step, is designed for post-training tasks where the distance between the original and trained parameters is small. Large distance of updates, such as pre-training, can make the update direction deviate too much from the direction of AdamW, leading to a slow training.

### E.2 ABLATION ON LEARNING RATE SCHEDULER

To evaluate how the learning rate scheduler introduced in Sec. 3.2.2, we conduct an experiment on Qwen2-0.5B, comparing AdamW without a scheduler and with a PMA-like scheduler. The other settings are the same as the experiment in Fig. 5. We evaluate the tuned model every 120 steps, and the statistics are shown in Fig. 10. The PMA-like scheduler slows down the training process if the other components of PMA are not applied. This result indicates the necessity of the joint design of each component in `AdamW-PMA`.

### E.3 SFT

The improvement in validation loss brought by PMA can be translated into a reduction of the number of steps or total compute. In Figure 11, we evaluate the optimizers by comparing the number of steps or total flops needed to achieve the same validation loss level, setting K to 4. As can be observed in Figure 12, AdamW-PMA and Lion-PMA achieve a 12x and 2x speedup compared with AdamW and Lion.

### E.4 DPO

Figure 13 and Figure 14 illustrate the validation loss of the DPO task on Phi-2 and HH-RLHF-harmless dataset, using four different optimizers. We compare the total flops and number of samples needed to achieve the same validation loss across vanilla AdamW and AdamW-PMA, Lion and

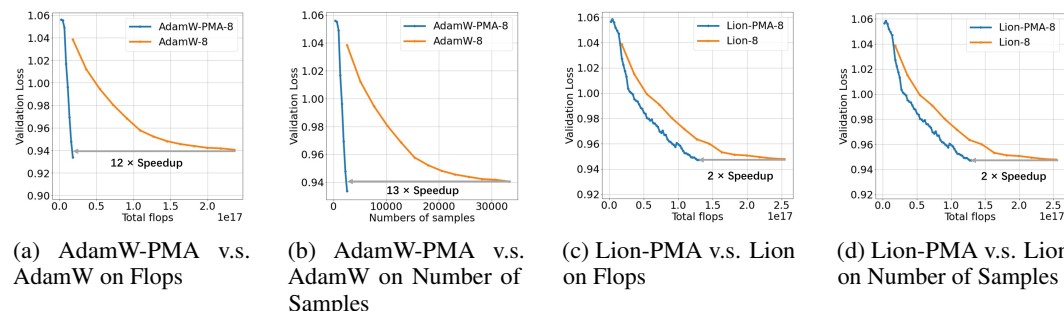

(a) AdamW-PMA v.s. AdamW on Flops

(b) AdamW-PMA v.s. AdamW on Number of Samples

(c) Lion-PMA v.s. Lion on Flops

(d) Lion-PMA v.s. Lion on Number of Samples

Figure 12: We evaluate the optimizers by comparing the total flops and number of samples needed to achieve the same validation loss level. `AdamW-PMA` and `Lion-PMA` achieved approximately 12x and 2x speedup, respectively, relative to AdamW and Lion.

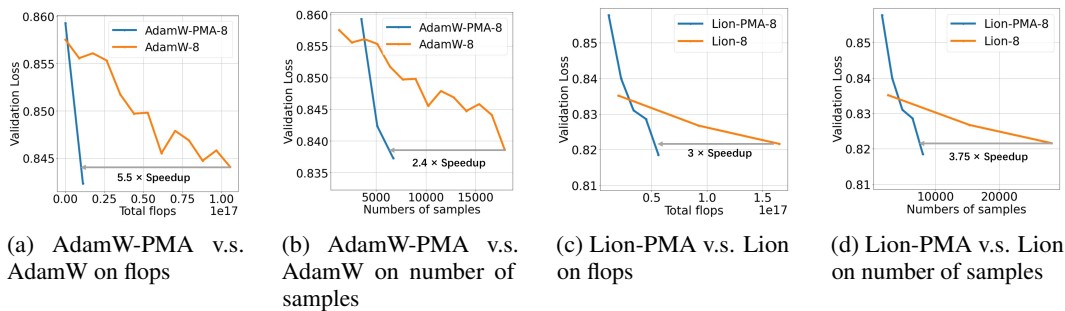

(a) AdamW-PMA v.s. AdamW on flops

(b) AdamW-PMA v.s. AdamW on number of samples

(c) Lion-PMA v.s. Lion on flops

(d) Lion-PMA v.s. Lion on number of samples

Figure 13: Validation loss of the DPO task on Phi-2 and HH-RLHF-harmless dataset.

Lion-PMA. The corresponding accuracy graph for this experiment can be found in Figure 3 of Section 5.3 in the main text.

### E.5 HYPER-PARAMETER SENSITIVITY

We do experiments on DPO task with the Phi-2-2.7B model and Qwen1half-0.5B-chat model to explore the sensitivity of the PMA method's speedup factor with hyper-parameter K on AdamW. In experiment of Phi-2 model, we set K to be 8, 16, 32, 64 to explore the optimal K value. For Qwen1.5-0.5B model, the K is set to be 4, 8, 16, 32, which are relatively smaller since the model is smaller. The results of experiments can be seen in Figure 15 and 16. This part is the supplement results of Section 5.4 in the main text.

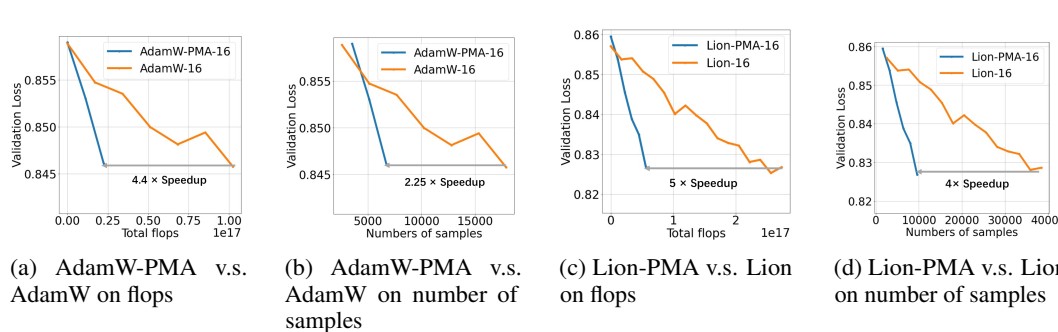

(a) AdamW-PMA v.s. AdamW on flops

(b) AdamW-PMA v.s. AdamW on number of samples

(c) Lion-PMA v.s. Lion on flops

(d) Lion-PMA v.s. Lion on number of samples

Figure 14: We evaluate the optimizers by comparing the total flops and number of samples needed to achieve the same DPO validation loss level, with K setting to be 16. `AdamW-PMA` and `Lion-PMA` achieved approximately 4x and 5x speedup, respectively, relative to AdamW and Lion.

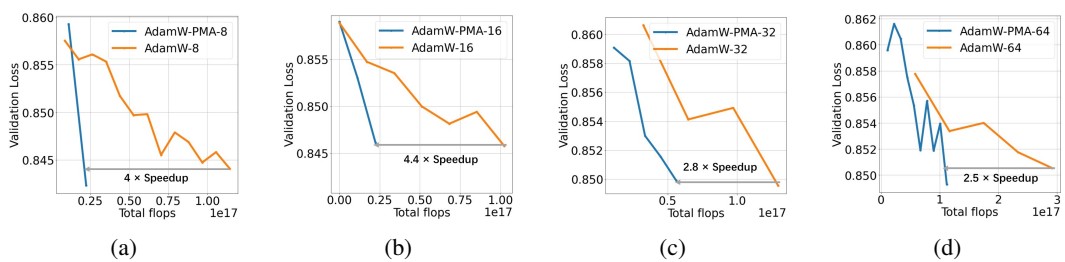

(a)     (b)     (c)     (d)

Figure 15: The sensitivity of PMA's speedup factor with hyper-parameter K on Phi-2 model using AdamW

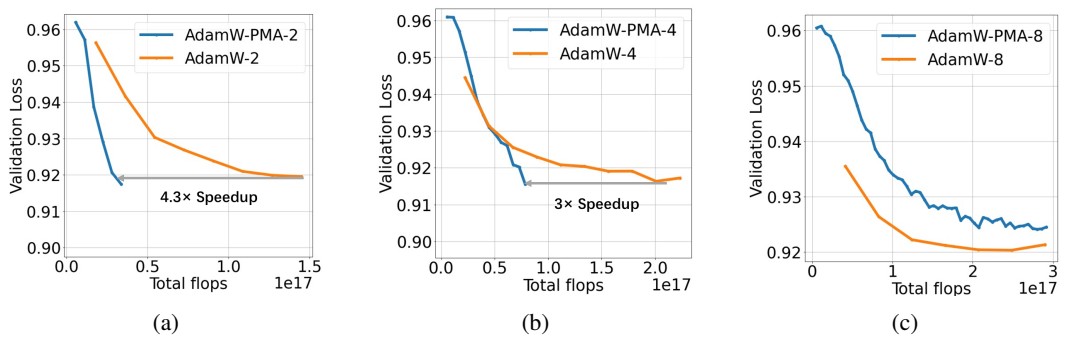

(a)     (b)     (c)

Figure 16: The sensitivity of PMA's speedup factor with hyper-parameter K on Qwenhalf1-0.5B model using AdamW

