# OpenReview forum: "Periodical Moving Average Accelerates Gradient Accumulation for Post-Training"
_ICLR.cc/2025/Conference — Submitted to ICLR 2025_

### Official Review · Reviewer_Ttrs · 2024-11-03

**Soundness:** 3
**Presentation:** 2
**Contribution:** 1
**Rating:** 3
**Confidence:** 4

**Summary:**

The paper proposes using an averaging style momentum for accelerating convergence of adaptive optimizers at small batch sizes. It provides experiments on SFT and DPO tasks, where it compares averaging style momentum at small batch sizes to EMA-style momentum at higher batch sizes and shows improved results.

**Strengths:**

It is an interesting idea to overlap EMA and averaging style momentum to reduce variance and this paper tries using it for improving convergence at small batch sizes.

**Weaknesses:**

1. For small batch sizes, a variation of momentum scheme known to accelerate convergence as described in these works [1,2]. This should be one of the baselines to compare the scheme to at small batch sizes. One easy way to implement this in Lion is simply to increase beta2 from 0.98 to much higher value while keeping beta1 around 0.95 or 0.9.

    [1] - Kidambi et al. 2018 - On the insufficiency of existing momentum schemes for Stochastic Optimization

    [2] - Gupta et al. 2023 - Achieving acceleration despite very noisy gradients

2. The baseline of AdamW at higher batch size does not seem fair as generally it is expected to have diminishing returns at higher batch sizes [1,2]. A better baseline would be to use normal AdamW at the same batch size.

    [1] - Shallue et al. 2018 - Measuring the Effects of Data Parallelism on Neural Network Training

    [2] - McCandlish et al. 2018 - An Empirical Model of Large-Batch Training

**Questions:**

I think the paper is currently missing very important baselines such as AdamW with normal momentum at the same batch size as well as the accelerated SGD variants of AdamW and Lion.

---

### Official Review · Reviewer_VRfj · 2024-11-04

**Soundness:** 3
**Presentation:** 3
**Contribution:** 3
**Rating:** 5
**Confidence:** 3

**Summary:**

The paper introduces PMA, a modification of the Gradient Accumulation (GA) process that incorporates updating the parameters when the accumulation is still in progress. During the accumulation a smaller learning rate is used for updating. The process can also be viewed as a periodic schedule for the decay hyperparameter in the momentum updates where $K$ consecutive steps of uniform averaging follows one step of exponential averaging.

The paper showed by experiments that adding PMA to existing optimizers boost validation accuracy across a series of tasks, and showed the effect of choice of $K$. Theoretically they showed that Adam+PMA enjoys a regret close to vanilla Adam when $K$ is small for convex optimization.

**Strengths:**

The paper has a clear illustration of their idea of PMA by adapting from GA. It includes comprehensive experiments to show the speed-up of effect in terms of FLOPs and sample numbers. The PMA algorithm is easy to implement and will be quite useful for boosting large-scale machine learning optimization algorithms.

**Weaknesses:**

1. The algorithms in the paper are confusing and seems to incorporate typos. For instance line 16-18 of algorithm 1 does not make any change to the optimization steps, and the scaling of $K$ in line 6-7 seems to be inconsistent to the text illustrations. The same is observed for algorithm 2. I don't know whether the following items stem from this confusion.

2. The current PMA modification of Adam implicitly alters the effective learning rate and decay hyperparameter of the original Adam algorithm: the previous momentum $m_{t-1}$ in 6 is counted multiple times in subsequent momentums in the period, so the same batch is added multiple times to the parameters, resulting in a larger effective learning rate and a different momentum decaying parameter beta. However in the experiments all the Adam and Adam-PMA uses the same literal learning rate and beta. Thus it is inconclusive whether the performance boost is due to the change in hyperparameters. The authors should show more experiments with different learning rates or should pick the optimal learning rate for Adam for start.

3. The hyperparameters in PMA (for instance Algorithm 1) seems not to be chosen optimally. For instance, the Debias step (12-13) uses a wrong constant for bias. Furthermore the learning rate in 9 is not optimally balancing the gradient variances: different $m_t$ in the same period actually have different variances due to the inhomogeneous batch sizes, and therefore different learning rates should be used. Furthermore the rational of dividing  gradient by $K$, rescaling momentum, etc. does not follow from a clear rationale.

The sensitivity of the algorithm performance w.r.t $K$ may be a result of these suboptimal choices.

4. The learning rate schedule in Figure 2 is not clear from text, as well as the meaning of the "actual learning rate" on line 273.

5. The theoretical analysis has limited implication. It can only show that Adam-PMA has a performance not too worse than Adam when $K$ is small, but cannot show any of its advantages in terms of noise reduction or additional updates. From a theoretical perspective, PMA should boost performance especially for large $K$ as the original GA uses too many steps in accumulation, yet this is not observed in the current analysis.

**Questions:**

1. Can you provide more clarification for weakness 1&3, namely a clearer mathematical rationale for the choices of the hyperparameters in algorithm 1?

2. Can you provide additional evidence for weakness 2, namely the performance boost is not due to implicitly changing the learning rates or momentum beta?

---

### Official Review · Reviewer_2K8J · 2024-11-04

**Soundness:** 3
**Presentation:** 3
**Contribution:** 3
**Rating:** 5
**Confidence:** 4

**Summary:**

Training Large Language Models (LLMs) on memory-constrained devices is challenging due to high gradient variance, which hampers efficient convergence. Common solutions, such as small batch sizes or Gradient Accumulation (GA), must balance low convergence rates from high parameter update variance with extended training times from GA's sequential process. This paper highlights a limitation in momentum updates: the Exponential Moving Average (EMA) quickly diminishes the influence of historical gradients, making it hard to stabilize update steps. To overcome this, the authors introduce the Periodical Moving Average (PMA) technique, which incorporates GA principles into momentum updates to better leverage historical gradients.

**Strengths:**

1. The high-level idea for the proposed PMA method is brilliant and highly well-motivated to me.
2. Experimental results with Adam-PMA/Lion-PMA variants show promising results compared to the baselines.
3. The convergence result (Theorem 1) helps to understand the proposed method.

**Weaknesses:**

I have several concerns.

1. Although the proposed PMA method seems quite novel and promising, the learning rate scheduler is also mandatory for plausible performance. However, in experimental sections, there is no ablation study for this, which I think is also important to evaluate the contributions of PMA scheme itself. I recommend that the authors conduct more comprehensive ablation study for the robustness.

---

2. While I appreciate the convergence results in Theorem 1, the analysis is an extension of convex regret analysis of original Adam. In this analysis, I could not see the benefit of variance reduction (large update steps) or small update steps. I think that such benefits should also be apparent in terms of theory, but it is missing in the current version.

---

3. In fact, this approach does not appear to be limited solely to LLM post-training. Gradient accumulation, for instance, is also used in memory-limited scenarios to perform large batch optimization. From this perspective, additional experiments in a broader range of domains—such as testing the method's effectiveness in areas like image classification or assessing its performance in large batch optimization scenarios—would make the findings more compelling (since some results seem to be marginal compared to the baselines).

**Questions:**

Please refer to the weaknesses.

---

### Official Review · Reviewer_2wLt · 2024-11-07

**Soundness:** 3
**Presentation:** 3
**Contribution:** 2
**Rating:** 5
**Confidence:** 3

**Summary:**

This paper proposes a Periodic Moving Average (PMA) to reduce the variance of AdamW, especially in the training of large language models. Instead of continuously doing EMA updates, the authors propose to only do EMA update every K steps, and within the inner loop, use the accumulated average of gradients to update. The proposed algorithm achieves more than 2 times the speedup, and better downstream tasks performance.

**Strengths:**

The authors propose a novel way to make use of the intermediate gradient calculations during the gradient accumulation, to also update the parameters, bringing better computational efficiency.

The authors also provided a theoretical justification on bounding the regret of the algorithm.

**Weaknesses:**

I would like to see more discussions on the relationship of this two-loop averaging mechanism, with traditional variance reduction algorithms. e.g., how did your algorithm reduces the variance, and how is it different with the variance reduction in SVRG?

The authors did not compare the variance of the EMA, with the variance of their proposed small step averaging iterates, making the claim of variance reduction confusing.

I think this paper is in general a good paper, however its relationship with previous algorithm are not thoroughly discussed. For example, I would like to see a comparison of the inner-loop updates with the Polyak–Ruppert Averaging.

**Questions:**

If we skip all the update steps in the small steps, is the algorithm equivalent to doing gradient accumulation and EMA updates? It would be better to clarify that, and if there are any differences, what is different?

---

### Meta-Review · Area_Chair_8ZT9 · 2024-12-19

**Metareview:**

Reviewers have raised several questions about how is the proposed approach compared to existing variance reduction methods and how the proposed approach is effective in variance reduction. There are also concerns about experimental settings and weak experiments. The authors of the paper did not provide any rebuttal. Hence, this is a rejection.

**Additional Comments On Reviewer Discussion:**

NA

---

### Decision · Program_Chairs · 2025-01-22

Reject